# Understanding Flood Vulnerability in Local Communities of Kogi State, Nigeria, Using an Index-Based Approach

**Peter Oyedele** [1,2,]*, **Edinam Kola** [1], **Felix Olorunfemi** [3] **and Yvonne Walz** [2]

1    West African Science Service Centre on Climate Change and Adapted Land Use (WASCAL) Graduate Research Program on Climate Change and Disaster Risk Management, Department of Geography, Université de Lomé, Lomé 01BP1515, Togo
2    Institute of Environmental and Human Security, United Nations University, Platz der Vereinten Nationen 1, 53113 Bonn, Germany
3    Nigerian Institute of Social and Economic Research, P.M.B 5, UI Post Office, Oyo Road, Ibadan 200132, Oyo State, Nigeria
*    Correspondence: bolupeter@gmail.com or oyedele.p@edu.wascal.org; Tel.: +234-703-655-3040

**Abstract:** In West Africa, the impacts of flooding are becoming more severe with climate warming. Flood-prone communities in Kogi State in north-central Nigeria are affected by annual flooding and some extreme flood events. The negative impacts remain a major obstacle to development, environmental sustainability, and human security, exacerbating poverty in the region. Reducing and managing the impacts of flooding are increasingly becoming a challenge for individual households. Analysing vulnerability to flooding (a function of exposure, susceptibility, and lack of resilience) and identifying its causes using an index-based approach to achieve sustainable flood risk management were the focus of this study. A semi-structured questionnaire was used to collect relevant data from 400 households in 20 purposively selected communities. Based on expert opinions and an extensive literature review, 16 sets of relevant indicators were developed. These indicators were normalised and aggregated to compute the flood vulnerability index (FVI) for each community. This was then used to compare, classify, and rank communities in terms of their vulnerability to flooding. The results of the study showed that the selected communities were at varying levels of the risk of flooding. Four of the communities including the Onyedega, Ogba Ojubo, Odogwu, and Ichala Edeke communities were found to have very high vulnerability to flooding compared to others. Several factors such as poor building structures, lack of evacuation and flood management measures, over-dependence of households on agriculture, lack of diversification of economic activities, and weak household economic capacity were identified as causes. These findings are useful for developing flood risk reduction and adaptation strategies, such as ecosystem-based approaches, to reduce current and future vulnerability to flooding in Nigeria and other developing countries with similar conditions.

**Keywords:** flood vulnerability; indicators; flood-prone communities; lack of resilience; Kogi State; Nigeria

## 1. Introduction

The frequency and severity of weather-related events such as floods are undoubtedly rising [1], due to the increasing risks associated with urbanization and the potential impacts of climate change [2]. Over the following decades, climate change is projected to have an increasingly negative impact on hydrological regimes and flood risks [3]. Floods continue to be one of the most frequently occurring and dangerous natural hazards, affecting human lives and resulting in significant economic losses around the world [4,5]. The recurrence of flooding events and the risk that goes along with them have a greater negative impact on developing countries [6] due to a variety of factors, including unstable economies, a lack of understanding of the hazard, inadequate preparation, and coping capacity [6,7]. Flood risk assessments and management are compulsory to determine the highest-risk areas in order to reduce the accompanied risk [8].

Nigeria has also recently experienced recurrent flooding that has cost lives and property [5]. According to the Nigeria National Emergency Management Agency (NEMA), Nigeria experienced one of its most devastating floods in 2012, affecting millions of people and resulting in financial losses of several billion USD [9]. As contained in the 2012 Post-Disaster Needs Assessment (PDNA) by the Federal Government of Nigeria, it was reported that the flood affected fourteen states, including Kogi, with devastating effects on the lives and property of households in flood-prone communities [10,11]. This event has now become a regular phenomenon, resulting in numerous casualties and losses.

In the last 10 years, about eight major floods have affected several communities in Kogi State. The most devastating were the floods in 1994, 2004, 2010, 2012, 2017, 2018, 2019, and 2020, which killed about 250 people, displaced 85,000 others, and caused several other damages worth millions of USD [9,12–14]. In particular, it was reported that about 150 communities across nine local government areas (LGAs) on the banks of the Niger and Benue rivers were submerged in floodwater during the September 2019 floods [13]. The negative impact of annual floods, as evidenced in the literature and media reports, remains a critical constraint to agricultural production, environmental development, food supply, and human security, thus deepening poverty in the region [15].

Considering how frequently and severely Kogi State is being flooded, various research works have been conducted in addressing various issues surrounding the causes, impacts, and management of flood disasters [15–19]. However, despite the huge contributions of these studies, up till now, we do not yet understand why the people are vulnerable and the various dimensions of these vulnerabilities. The understanding of factors that trigger people's underlying vulnerabilities and the negative consequences is necessary for managing flood risks efficiently [20]. Decision-makers and the research community now acknowledge vulnerability assessment as a necessity for developing successful risk, flood vulnerability reduction measures, as well as a critical requirement for understanding society's exposure to environmental hazards [21–23]. This informed the need to investigate why people are affected by floods in terms of their exposure and lack of resilience.

It is commonly agreed that three interrelated factors interact to generate flood risk: the flood hazard, the exposure of people and property, and the susceptibility of exposed populations and buildings to flood impacts [24]. Furthermore, reducing the exposure and vulnerability of people, property, infrastructure, and other assets to floods remains a significant goal of flood risk management [24,25]. Nazeer and Bork [26] noted that one of the often-employed techniques for assessing flood vulnerability is an empirical investigation using flood vulnerability composite indicators. According to Quesada-Román [8], an index for flood risk was designed to comprehend the risk drivers' role (hazard, exposure, and vulnerability). Balica et al. [27] avowed that the Flood Vulnerability Index (FVI) remains an effective instrument for policymakers to prioritize investments that increase transparency in the decision-making process in their quest to address flood risk management. The use of the FVI will help to identify the places most at risk of flooding and the drivers, with informed decisions on areas to be considered in future redevelopments [28]. As demonstrated by the literature review above, research on and understanding of flood-related vulnerability in this area are still lacking, which motivated this study to address this gap with a focus on households in the flood-prone communities of Kogi State, Nigeria.

Taking these concerns into consideration, the aim of this study is to analyse the vulnerability of selected areas to flooding, identify geographically the hotspots of flood vulnerability in the area, and identify the factors that influence the vulnerability using the Flood Vulnerability Index (FVI) approach. We assumed that all households in the sampled communities have the same factors influencing their vulnerability to flooding and, hence, a similar flood vulnerability status. The FVI is indeed an effective tool for locating regions that are highly prone to floods and also for assisting in the decision-making to improve flood management in the selected communities. The resulting index value will assist in identifying communities that are most at risk to flooding and the factors that contribute to this risk. This method is capable of guiding experts in disaster risk management and

relevant authorities in the implementation of appropriate location-specific solutions in the form of adaption and mitigation measures. Undoubtedly, the success will motivate researchers not only in Nigeria, but also other developing countries.

## 2. Materials and Methods

### 2.1. The Study Area

The study was carried out in Kogi, Nigeria, located between latitudes 7°301 N and 7°521 N and longitudes 6°381 E and 6°421 E. It is one of the states in the north-central geo-political zones of Nigeria with a total land area of 25,934 sq. km and a projected population of 4,473,500 in 2021 (https://kogistate.gov.ng/structure/, accessed on 8 November 2021). Two major drainage systems flow, the Niger and Benue Rivers, forming a confluence in Lokoja, the state capital. The convergence of these two rivers makes the state one of the most flooded in the country [29]. It has twenty-one (21) local government areas (LGAs). Eight of the LGAs were selected for the study based on the fact that they are the most flooded [9,13]. These are Ibaji, Koton-Karfe, Lokoja, Ofu, Ajaokuta, Omala, Bassa, and Idah (Figure 1).

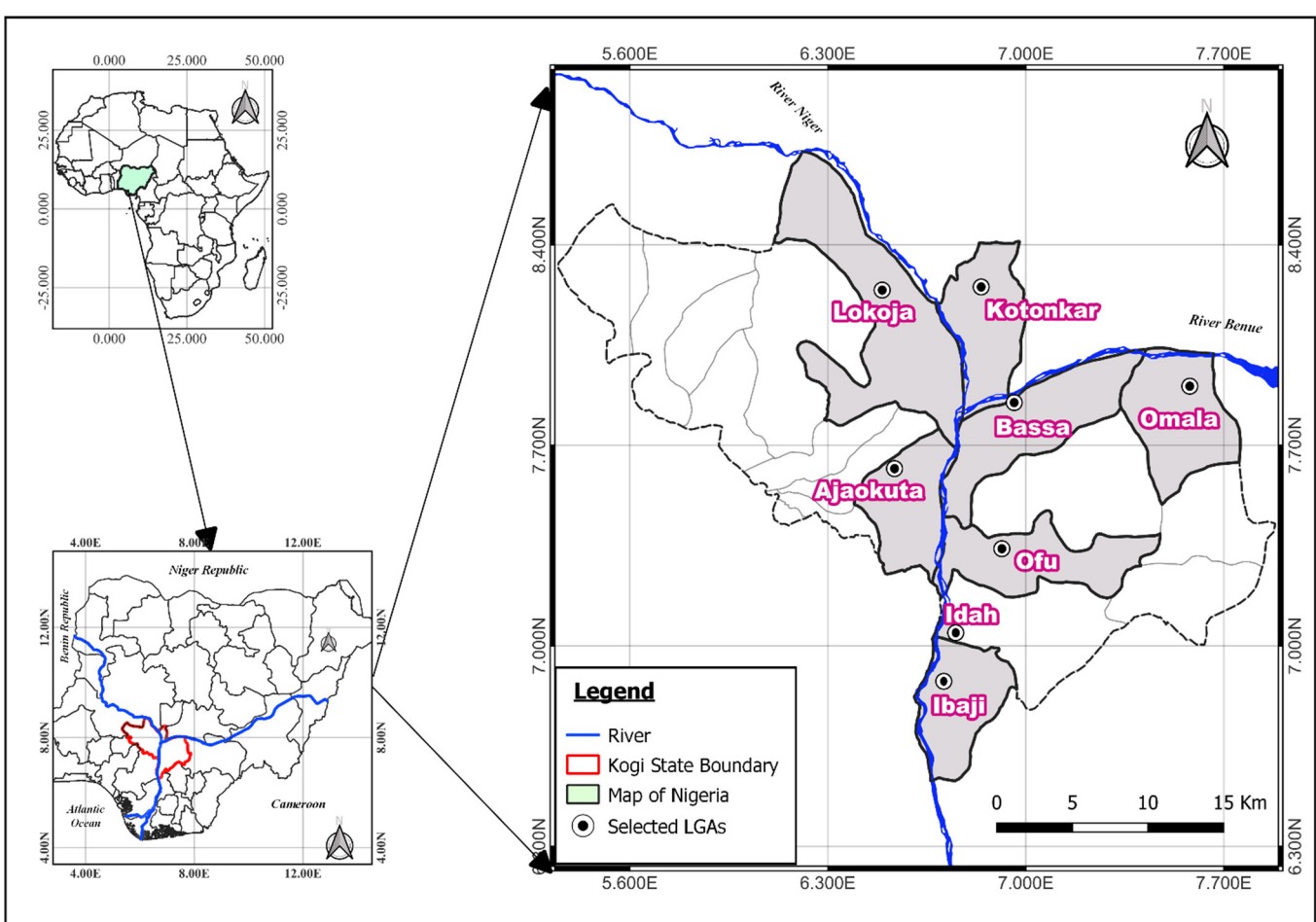

**Figure 1.** Study area map.

Communities in the LGAs were also purposively selected for this study. The reason for choosing these locations was that about 150 communities in these LGAs situated along the Niger and Benue Rivers were partially or totally submerged in floodwater during a recent devasting flooding event, as reported by Pulse.ng [13]. Additionally, the communities near the channels and at the confluence of these rivers suffer the full effects of flood occurrences [30].

## 2.2. Data Collection, Sampling, and Questionnaire Design

The data used in this study were obtained from both primary and secondary sources. Most of the primary data came from field studies that included stakeholder meetings, reconnaissance, questionnaire surveys, field observations, and interviews. Between October 2020 and January 2021, the stakeholder meetings, reconnaissance, and ground truthing were completed. Interviews were conducted with local leaders, Kogi State Emergency Management Agency (SEMA) staff and officials, and the Director of Ministries (Environment and Agriculture, respectively) to obtain more information on past flooding. Additionally, focus group discussions (FGDs) were held at four different locations. The relative extent of flood vulnerability of some large-scale units is best determined using secondary data [31]. Information on past flood events in the area was gathered from the literature, government agency reports, and documents. A base map of Kogi State covering the selected LGAs was produced using tools from GIS. From the National Space Research and Development Agency (NASRDA) website, the rivers maps were downloaded.

The study's respondents were selected using a multi-stage sampling procedure. Eight LGAs that were severely impacted by the 2019 flood in terms of the number of persons displaced and economic loss were selected. Three communities were randomly selected from the Lokoja, Kogi-Koto Karfe, Bassa, and Ibaji LGAs, while two communities were selected from the Omala, Ajaokuta, Ofu, and Idah LGAs. This brought the total number of communities selected to twenty (Table 1). Due to constraints in the mobility, availability, and accessibility of respondents, only 20 respondents were purposively selected from each of the selected communities. Semi-structured questionnaires containing relevant indicators were used in the collection of the data from 400 farming households (which included either the father, mother, or adult child) between March and June 2021.

**Table 1.** Sampled size of households in the selected communities.

| Local Government Area | Community | Sampled Households |
|---|---|---|
| Ajaokuta | Geregu and Adogu | 40 |
| Bassa | Eroko, Icheu, and Shintaku | 60 |
| Ibaji | Odogwu, Ogba Ojubo, and Onyedaga | 60 |
| Idah | Ichekene and Ichala Edeke | 40 |
| Koto Karfe | Edeha, Apaku, and Koto karfe | 60 |
| Lokoja | Kakanda Budon, Adankolo, and Karara | 60 |
| Ofu | Itobe and Olukudu | 40 |
| Omala | Bagana and Abejukolo | 40 |

## 2.3. Data Collection, Sampling, and Questionnaire Design

The construction of flood vulnerability indicators as performed in this study built on several studies [26,32–34] that developed flood vulnerability composite indices, which generally followed the Organization for Economic Cooperation and Development's methodology [32] multi-step workflow, as modified in (Figure 2). This includes: (i) indicator derivation, (ii) normalization of indicators and characterizing the indicator, (iii) weighting of normalized indicators, (iv) aggregation of weighed indicators, and (v) flood vulnerability mapping.

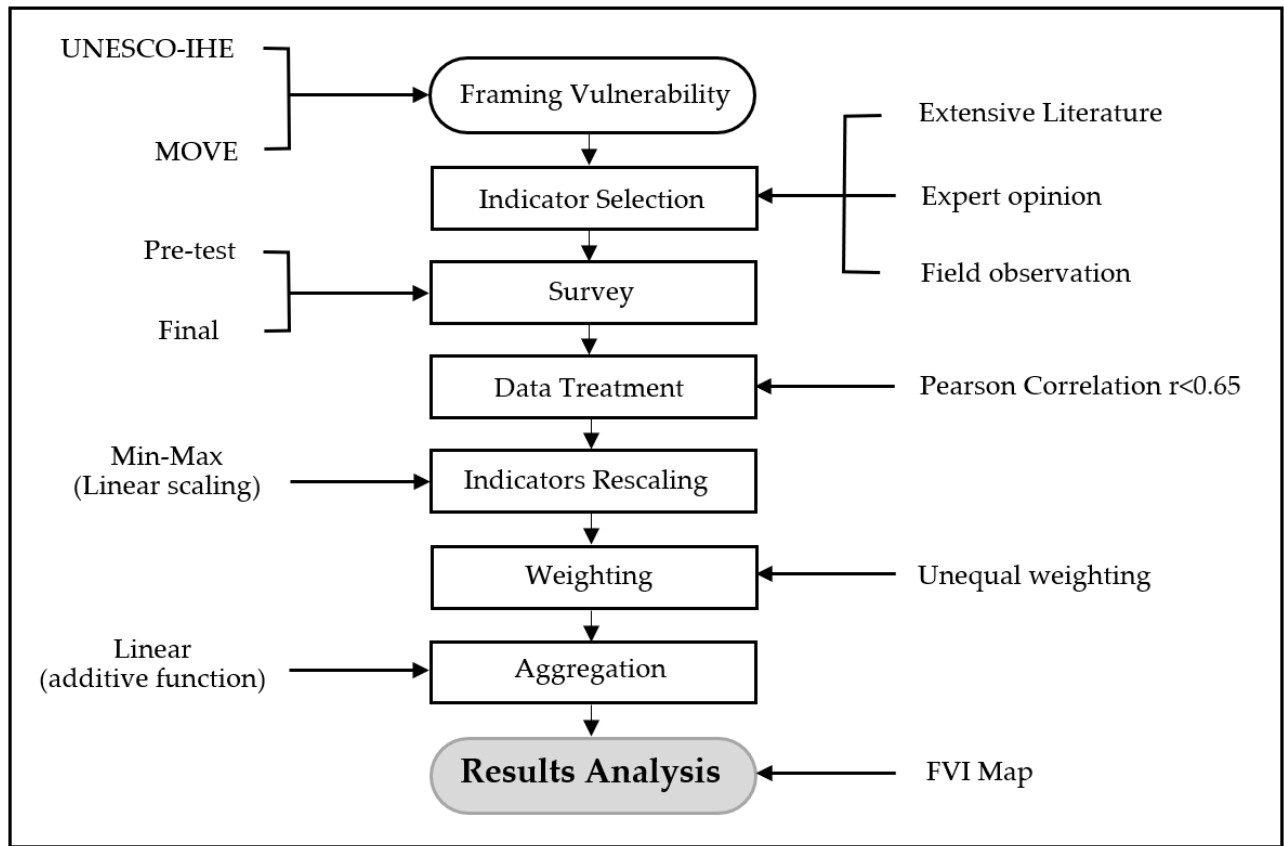

**Figure 2.** Flood vulnerability indices' development workflow. Adapted from [26].

### 2.3.1. Framing Vulnerability and Description of Vulnerability Indicators

By utilizing the UNESCO-IHE (Institute of Water Education, Delft, The Netherlands) and the Methods for the Improvement of Vulnerability Assessment in Europe (MOVE) frameworks for the description of flood vulnerability, this study adopted deductive reasoning for the preliminary set of indicators' selection [35]. In contrast to how UNESCO-IHE defines vulnerability as the combination of exposure, susceptibility, and resilience components, the MOVE framework maintains the negative definition of vulnerability and alludes to "lack of resilience" rather than just "resilience."

*Exposure (E):* This explains the degree to which a region that is the focus of an assessment falls within the scope of a hazardous event [35]. It refers to the possibility that flooding will have an effect on individuals, as well as possible tangible items (properties, buildings, cultural heritage, and agricultural land) because of their position [36].

*Susceptibility (S):* defines the propensity of elements at risk (social and ecological) to suffer harm as a result of the level of settlement volatility, unfavourable conditions, and relative weaknesses [35,37].

*Lack of resilience (LoR):* This means the inabilities to anticipate, cope with, and recover from the effect of a natural hazard. It comprises pre-event risk reduction, in-time coping, and post-event response actions [35]. Similar to this, it highlights the socio-ecological system's restrictions to resource access and mobilization, as well as its inability to respond by absorbing the damage [38].

### 2.3.2. Indicators' Derivation

Most vulnerability analysis is based on indicator selection and analysis [26,39]. Adger and Vincent [40] advocated for the usefulness, appropriateness, data availability, and ease of recollection of indicators in vulnerability assessment. A review of the literature helped in understanding the different types of indicators used in the vulnerability analysis and

mapping around the world. Here, a list of these indicators was made and documented. This was complemented by an empirical observation from the field. The dataset used in defining each of the vulnerability components (exposure, susceptibility, and lack of resilience) were empirically analysed using appropriate statistical methods, as explained under the methodology to derive indicators that best defined the individual component of vulnerability in the study area. The list of indicators from the literature and empirical field observation were presented before a team of experts comprising five staff of the Kogi State Ministry of Environment (Climate Change Unit), three members of the Kogi State Ministry of Agriculture, four experts from the National Inland Waterways Authority (NIWA) in Kogi State, and three members of the Kogi State Emergency Management Agency (SEMA). With the intervention of these experts, some indicators were retained, while others were deleted based on their opinions. In the end, a non-exhaustive list of 18 indicators was derived (Table 2).

**Table 2.** Flood vulnerability indicators of flood-prone communities of Kogi State and their functional relationship.

| Vulnerability Components | Indicators (Units) | Abbr. | Justification/Explanations | Functional Relationship (+/−) | References |
|---|---|---|---|---|---|
| Exposure (E) | Average elevation (m) | AE | Flood exposure increases with decreasing elevation, hence the higher the vulnerability | (+) | [41–43] |
| | Closeness of farmlands to river bodies (m) | CRB | The closer the farmlands are to active water channels, the higher is the vulnerability | (+) | [42–44] |
| | Floodwater duration (days) | FD | The longer the floodwater persists, the higher the vulnerability | (+) | [42,44] |
| | Share of exposed farmland (%) | SEF | The higher the % of farmland, the higher the potential of flood exposure and the higher the vulnerability | (+) | [45] |
| Susceptibility (S) | Household size (avg.) | HS | The higher the avg. number of household size, the more the dependency rate, the higher the people's susceptibility, and the greater the vulnerability | (+) | [46,47] |
| | House conditions: number of houses with poor material (Avg.) | HCs | The more the number of houses with poor building materials, the higher the susceptibility, hence the more vulnerable, the higher the vulnerability | (+) | [44,46,47] |
| | Past flood experience (%) | PFE | The less flood experience people have, the more they are susceptible to become affected and the higher the vulnerability | (+) | [44] |
| | Household's dependency on agricultural production (%) | HDAP | The more the % of household dependency on agricultural production, the higher the susceptibility of affected people to be affected by flooding and the higher the vulnerability | (+) | [31,48] |
| | Lack of access to improved drinking water (%) | LAIW | The higher the % of people with a lack of access to improved drinking water, the higher the susceptibility of the affected people and the higher the vulnerability | (+) | [31] |

**Table 2.** *Cont.*

| Vulnerability Components | Indicators (Units) | Abbr. | Justification/Explanations | Functional Relationship (+/−) | References |
|---|---|---|---|---|---|
| Lack of Resilience (LoR) | Literacy rate: percentage of population with higher education (%) | LR | The higher the literacy rate, the more their capacity to anticipate, hence the lower people's vulnerability | (−) | [31,37,48] |
| | Access to Flood warning system/facilities/information (%) | AFWS | The higher the %, the higher the capacity to anticipate, hence the lower people's vulnerability | (−) | [44,49,50] |
| | Flood education (training) access rate (%) | FEAR | The higher the access rate to training on floods, the higher the people's capacity to anticipate for flooding and the lower the vulnerability | (−) | [47,51] |
| | Means of evacuation facilities (%) | MEF | The higher the % of households that have the ability to evacuate when a flood disaster strike, the more their capacity to cope and the lower the vulnerability | (−) | [35,47,49,51] |
| | Long-term residents at least 10 years + (%) | LTR | The higher the %, the longer the household settled in flood-prone areas, the more experienced they are, the higher their ability to cope, and the lower the vulnerability | (−) | [41] |
| | Access to healthcare and social services (%) | AHS | The higher the %, the more the ability of the affected population to cope and the lower the vulnerability | (−) | [31] |
| | Access to financial aid to face flood disasters (%) | AFA | The higher the % of household with access to financial and social assistance, the higher the capacity to cope and the lower the vulnerability | (−) | [42] |
| | Access to flood management measures (%) | AFMM | The higher the % of household with access to flood management measures, the higher the capacity to recover and the lower the vulnerability | (−) | [42] |

### 2.3.3. Data Treatment

According to Damm [52], a high degree of the linear relationship between indicators may distort the vulnerability index and mislead the end users. Therefore, to avoid the loss of important information, the redundancy of indicators, and a misleading vulnerability index in the end, the data obtained were subjected to treatment prior to data rescaling, weighting, and aggregation. Since all the indicators were quantitative in nature, the study adapted the approach of Damm [52] to determine the relationship among the indicators using the Pearson correlation. In the analysis, two or more highly correlated indicators with more than a 65% ($r > 0.65$) relationship were analysed to consider the removal of one of them.

### 2.3.4. Normalization of Indicator

The indicators obtained come with different units and scales. To have a comparable set of indicators, the study adopted the Min–Max normalization to convert the values to a linear scale (such as 0 to 1) [26,37,53].

There are two distinct forms of functional relationships to take into consideration:

(a)    Vulnerability (V) increases as the absolute value of the indicator also increases. In this case, where the functional relationship between the indicator and vulnerability is positive, the normalized indicator is derived using the following equation:

$$X_i = \frac{X_a - X_{Min}}{X_{Max} - X_{Min}} \tag{1}$$

(b)    Vulnerability (V) decreases with an increasing absolute value of the indicator. Here, when the relationship between vulnerability and the indicator is found to be negative, the data are rescaled by applying the equation below:

$$X_i = \frac{X_{Max} - X_a}{X_{Max} - X_{Min}} \tag{2}$$

where:

$X_i$ = normalized value;
$X_a$ = actual value;
$X_{Max}$ = maximum value;
$X_{Min}$ = minimum value for an indicator $i$ (1, 2, 3, . . . , $n$) across the selected communities.

### 2.3.5. Weighting of Indicator

No weight was assigned to the indicators. The reason for not including weights was that most responses during the stakeholders' engagement were contradictory and highly conflicting. Therefore, to avoid an index value that will mislead the end users, the normalized indicator was aggregated into its respective sub-indices for the final flood vulnerability index [26].

### 2.3.6. Aggregation of Indicator

The additive arithmetic function was employed in the aggregation of the indicator into its respective sub-indices (exposure, susceptibility, and lack of resilience) using Equation (3) [26,31,41]:

$$SI = \frac{\sum_{i=1}^{n} X_i}{n} \tag{3}$$

The overall flood value of the vulnerability index was computed with Equation (4), an additive function [31,54]:

$$FVI = \frac{1}{3} \left( SIE + SIS + SLoR \right) \tag{4}$$

where *SI* means sub-indices exposure (*SIE*), susceptibility (*SIS*), and lack of resilience (*SILoR*) for "*n*" numbers of indicator in each component of vulnerability.

### 2.4. Data Analysis

For statistical analysis, the questionnaire survey data collected were subjected to several statistical analyses: First, a data code sheet was developed and used to uniformly code the data for entry purposes using EpiData version 3.1. Applying Equations (1)–(4), the calculated vulnerability index value ranges from 0 to 1, with 1 denoting the highest vulnerability and 0 signifying no vulnerability at all. In Table 3, using an equal-interval method, the obtained FVI values were grouped into five classes following Kablan et al. [37].

**Table 3.** Flood vulnerability index ranking for selected flood-prone communities in Kogi State.

| Index Value | Description | Designated Colour |
|---|---|---|
| 0.32–0.40 | Very low vulnerability | Light Green |
| 0.40–0.48 | Low vulnerability | Dark Green |
| 0.48–0. 57 | Moderate vulnerability | Yellow |
| 0.57–0.65 | High vulnerability | Orange |
| 0.65–0.74 | Very high vulnerability | Red |

Adapted from [44].

## 3. Results

This section presents the findings from the research by first describing the hotspots of flood vulnerability, as well as understanding the drivers of flood vulnerability across the selected communities in Kogi State.

### 3.1. Identification of the Study Area's Flood Vulnerability Hotspots

To further achieve the set objectives for this study, efforts were made to determine and classify the communities under study into areas where there exists a prevalence of flooding events with respect to the acquired and analysed data. The FVI and other sub-indices values were computed, ranked, and categorized to determine the hotspots of flood vulnerability in the area.

#### 3.1.1. Flood Vulnerability Index

Applying Equations (1)–(4), the computed values for exposure, susceptibility, lack of resilience, and the overall FVI are presented in Table 4. From the table, the FVI values lie between 0.32 and 0.74, while the sub-indices values of exposure, susceptibility, and lack of resilience were (0.17–0.87), (0.28–0.83), and (0.38–0.82) respectively. This shows there were considerable variations in tract-level flood vulnerability and its three components across the selected communities.

**Table 4.** Kogi State flood-prone communities' Flood Vulnerability Indices.

| Selected Community | Sub-Index Exposure (*SIE*) | Sub-Index Susceptibility (*SIS*) | Sub-Index Lack Resilience (*SILoR*) | Flood Vulnerability Index (*FVI*) |
|---|---|---|---|---|
| Shintaku | 0.29 | 0.28 | 0.38 | 0.32 |
| Ichekene | 0.17 | 0.73 | 0.56 | 0.48 |
| Geregu | 0.37 | 0.42 | 0.71 | 0.50 |
| Abejukolo | 0.33 | 0.65 | 0.63 | 0.54 |
| Eroko | 0.45 | 0.68 | 0.49 | 0.54 |
| Bagana | 0.33 | 0.60 | 0.70 | 0.55 |
| Olukudu | 0.59 | 0.60 | 0.51 | 0.57 |
| Kakanda | 0.48 | 0.68 | 0.60 | 0.59 |
| Adankolo | 0.46 | 0.75 | 0.60 | 0.60 |
| Adogo | 0.53 | 0.74 | 0.55 | 0.61 |
| Adaha | 0.60 | 0.83 | 0.43 | 0.62 |
| Itobe | 0.64 | 0.77 | 0.50 | 0.64 |
| Icheu | 0.55 | 0.79 | 0.58 | 0.64 |
| Karara | 0.72 | 0.72 | 0.51 | 0.65 |
| Akpaku | 0.79 | 0.73 | 0.48 | 0.67 |
| Koton karfee | 0.87 | 0.65 | 0.50 | 0.67 |
| Ichala Edeke | 0.47 | 1.00 | 0.64 | 0.70 |
| Ogba Ojubo | 0.61 | 0.73 | 0.82 | 0.72 |
| Onyedega | 0.73 | 0.76 | 0.69 | 0.73 |
| Odogwu | 0.68 | 0.74 | 0.82 | 0.74 |

### 3.1.2. Ranking of the Communities Based on FVI and Other Sub-Indices' Values

Using these computed FVI and other sub-indices' values, the selected communities were ranked following [55] (see Table 5). Based on the FVI in particular, the Odogwu and Shintaku communities were ranked the highest and lowest, respectively. With respect to the exposure level, Koton-Karfe and Ichekene were found to be the highest and lowest exposed communities, respectively. In the same vein, considering the sub-index susceptibility, Ichala Edeke was found to be the community with the highest flood susceptibility, while Shintaku ranked as least comparatively. In relation to a lack of resilience, households in the Ogba Ojubo and Odogwu communities were both ranked first in the prevailing characteristics of a higher lack of resilience accordingly. In contrast, Shintaku on the other hand, was ranked as the community with the lowes lack of resilience to flooding. The study showed that the first three ranked communities are from the Ibaji local government area.

**Table 5.** Ranking of the flood-prone communities based on their FVI values.

| LGAs | Communities | FVI | Rank Based on | | | |
|---|---|---|---|---|---|---|
| | | | FVI | SIE | SIS | SILoR |
| Ibaji | Odogwu | | 1 | 5 | 8 | 2 |
| Ibaji | Onyedaga | | 2 | 3 | 5 | 5 |
| Ibaji | Ogba Ojubo | | 3 | 7 | 10 | 1 |
| Idah | Ichala Edeke | | 4 | 13 | 1 | 6 |
| Kogi Koto | Koton karfe | | 5 | 1 | 16 | 15 |
| Kogi Koto | Akpaku | | 6 | 2 | 9 | 18 |
| Lokoja | Karara | | 7 | 4 | 12 | 13 |
| Bassa | Icheu | | 8 | 10 | 3 | 10 |
| Ofu | Itobe | | 9 | 6 | 4 | 16 |
| Kogi Koto | Adaha | | 10 | 8 | 2 | 19 |
| Ajaokuta | Adogo | | 11 | 11 | 7 | 12 |
| Lokoja | Adankolo | | 12 | 14 | 6 | 8 |
| Lokoja | Kakanda | | 13 | 12 | 14 | 9 |
| Ofu | Olukudu | | 14 | 9 | 17 | 14 |
| Omala | Bagana | | 15 | 18 | 18 | 4 |
| Bassa | Eroko | | 16 | 15 | 13 | 17 |
| Omala | Abejukolo | | 17 | 17 | 15 | 7 |
| Ajaokuta | Geregu | | 18 | 16 | 19 | 3 |
| Idah | Ichekene | | 19 | 20 | 11 | 11 |
| Bassa | Shintaku | | 20 | 19 | 20 | 20 |

Note: The FVI value is represented by the length of the bar in each cell. Low rank values (1, 2, 3, . . . ) for the FVI, SIE, SIS, and SILoR indicate higher flood vulnerability, higher exposure, higher susceptibility, and a higher lack of resilience, correspondingly and conversely at a relative level.

### 3.1.3. Categorization of Selected Communities Based on Flood Vulnerability Indices' Values

Following Kablan et al. [37], the ranked communities were further categorized into five subcategories, with a 0.74 FVI value considered as very high flood vulnerability and 0.32 indicating very low flood vulnerability. The result is presented in Figure 3, where each community falls into at least one of the categories. It was observed that almost 20% (four) of the communities were designated in red colour (very highly vulnerable, red colour). Similarly, highly vulnerable (orange) communities accounted for 50% (10), and 25% (five) were identified to be moderately vulnerable to flooding in the area. This implies that more than two-thirds of the surveyed communities are highly vulnerable to flooding and its negative impacts. This suggests that the majority of the sampled households face the impact of flooding in the sampled communities. In addition, the result equally showed that 90% of the very highly vulnerable communities (Ogba Ojubo, Onyedega, Odogwu) are

from the Ibaji LGA. This implies that, in relative terms, the Ibaji LGA is the most vulnerable to flooding among the sampled LGAs.

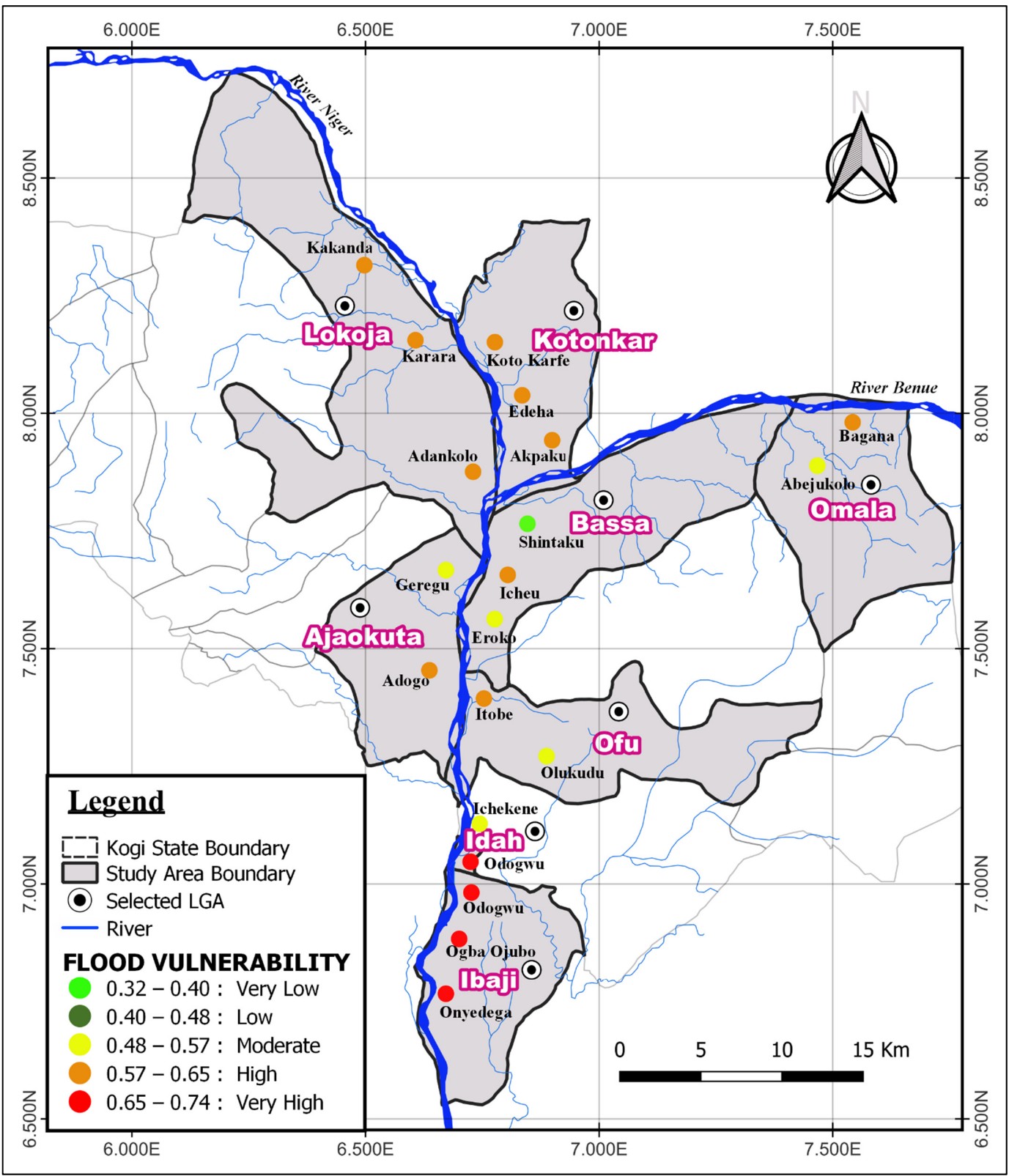

**Figure 3.** Hotspots of flood vulnerability among the selected communities in Kogi State, Nigeria.

### 3.2. Understanding the Drivers of Flood Vulnerability among the Selected Communities

To better inform decision-makers and professionals on the underlying causes of flood vulnerability in the area of study, the contribution of the sub-indices of exposure, susceptibility, and lack of resilience, SIE, SIS, and SILoR, respectively, to the FVI was evaluated. Similarly, efforts were made to clarify the contribution of the single indicator in each of the vulnerability sub-indices across the community.

3.2.1. Drivers of FVI and Its Underlying Factors in the Study Area

Figure 4 shows the contributions of vulnerability sub-indices to the prevailing levels of households' flood vulnerability in the communities studied. In the graph, it was clear that the sub-index susceptibility contributed most to flood vulnerability, followed by lack of resilience and exposure, in that order.

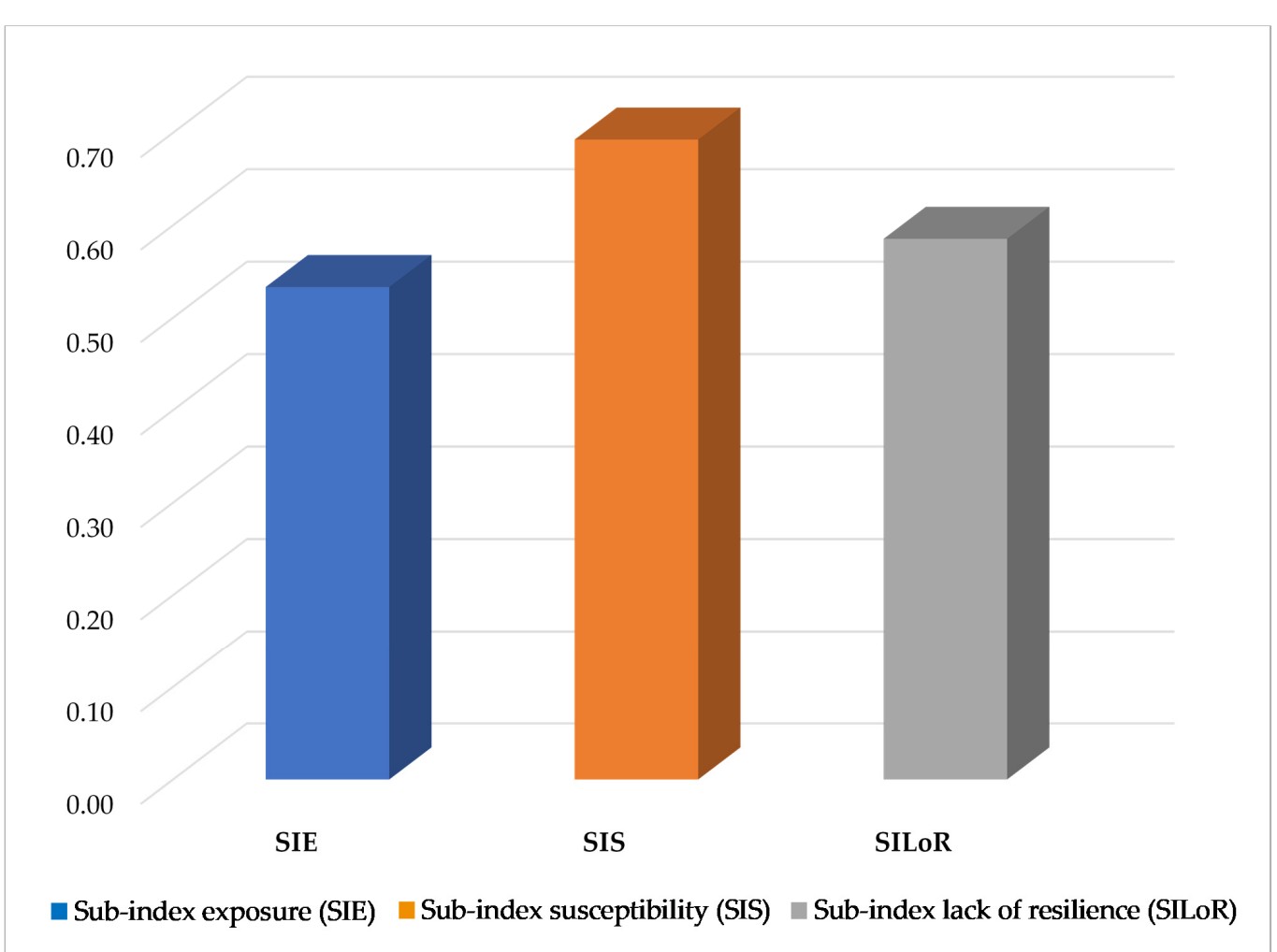

**Figure 4.** Contribution of vulnerability sub-indices in the FVI across the study area.

To provide practitioners and decision-makers a deeper understanding of the underlying factors influencing households' flood vulnerability in the areas under study, the contributions of the indicators selected for each vulnerability component were further evaluated. To this end, certain indicators were found to "push up" the FVI value due to either high exposure, high susceptibility, and/or a high lack of resilience, which we designate as "drivers" of vulnerability. On the contrary, variables that "pull-down" flood vulnerability levels due to either low exposure, low susceptibility, and/or low lack of resilience in a given area were considered as "buffers". Following Krishnan et al. [55], a sunburst plot was used to show the general contribution of individual indicator to flood vulnerability status

(Figure 5). Nine drivers influencing high vulnerability in the study area were household past flood experience (PFE), household dependency on agriculture (HDPA), not having access to improved and portable drinking water (LAIW), house conditions (HCs), access to flood management measures (AFMM), flood education access rate (FEAR), access to the healthcare system (AHS), access to financial aid (AFA), a low literacy rate, share of exposed farmland (SEF), and floodwater duration (FD).

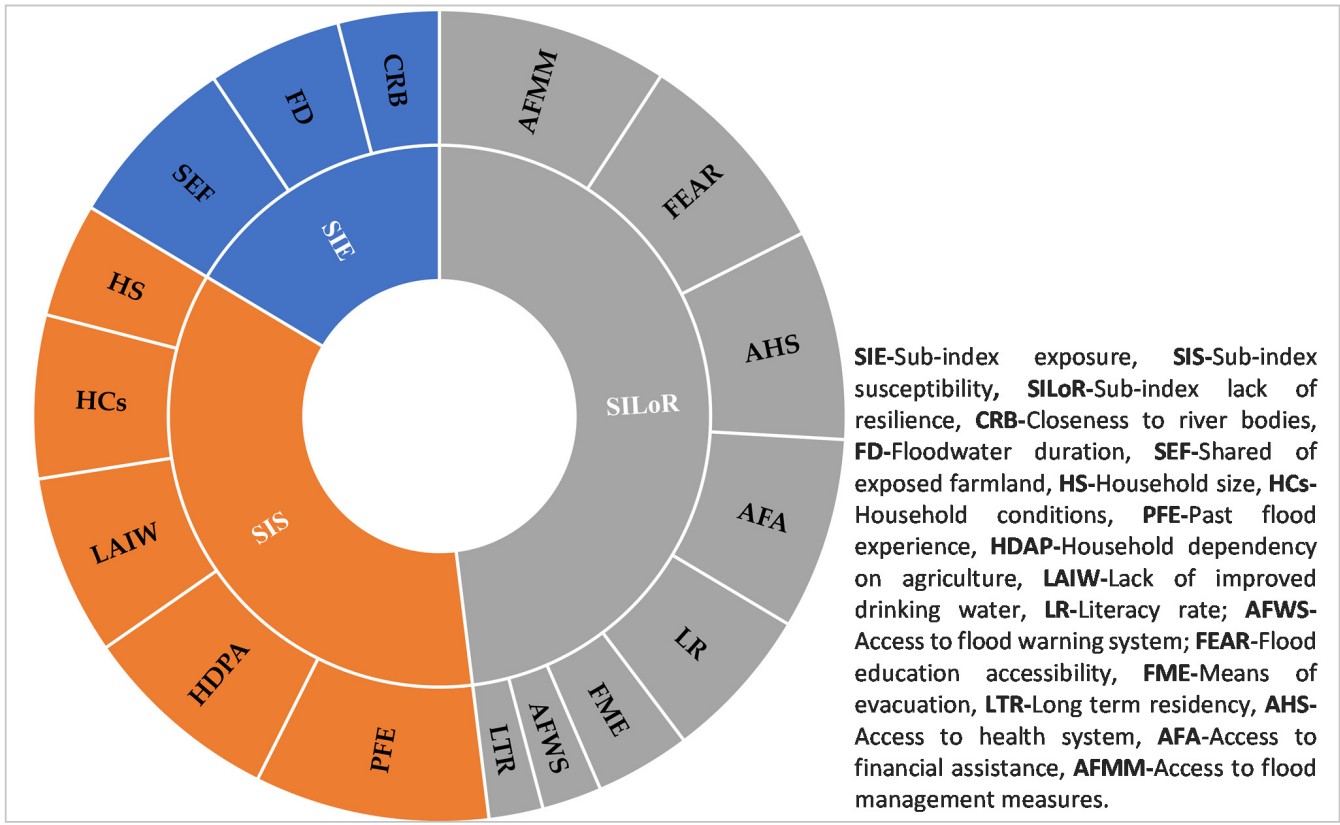

**Figure 5.** Contributors in the flood vulnerability index in the study area for intervention planning.

It is evident from the foregoing that high vulnerability is structural, in part due to the obvious predominately agrarian economy and unpleasant memories from previous flood events, largely defined by a relative lack of access to financial assistance, leading to a high percentage of flooded farmland. Included also is not having access to clean water and hygiene, which is further exacerbated by a lack of flood education rate and poor accessibility to the healthcare system. The key buffers that stabilised "vulnerability" were long-term residents at least 10 years + (%) (LTR), access to a flood warning system (AFWS), and means of evacuation facilities (MEF).

### 3.2.2. Contributions of the Single Indicator to the Sub-Indices Value

The sub-index exposure, susceptibility, and lack of resilience were further subjected to analyses at the community level. The results showed the percentage contributions of each indicator as it influences the community's vulnerability to flooding. This analysis was considered germane in order to critically understand the indicator that drives or influence each component of vulnerability, and secondly, to develop spatial contingency plans that allow for prompt response in the event of a flood disaster and promote resilience building:

(a)     Contribution of the single indicator to the sub-index exposure (SIE) across the community.

Three indicators were selected for the development of sub-index exposure. Following the approach of Hagenlocher and Castro [56], Figure 6 shows the contributions of these indicators to flood exposure across the communities. From the graph, it can be seen

that all the three indicators contributed to prevailing levels of flood exposure among the communities: (1) share of exposed farmland, (2) closeness to river bodies, and (3) floodwater duration. In addition, the analysis of the household survey conducted showed that more than half of the respondents (73%) engaged in farming and other forms of agricultural activity. First, this implies that farming is an important economic activity and a major aspect of the livelihoods among the people. It equally suggests the likelihood of the households' farmlands being impacted during flooding. Moreover, many (78.8%) of the respondents revealed that it takes up to forty-five days for floodwater to dry up in their neighbourhoods. In Koton-Karfe in particular, the three indicators have a similar percentage contribution to flood exposure; this generally accounts for the reason why the community had the highest sub-index exposure value (0.87) compared to others.

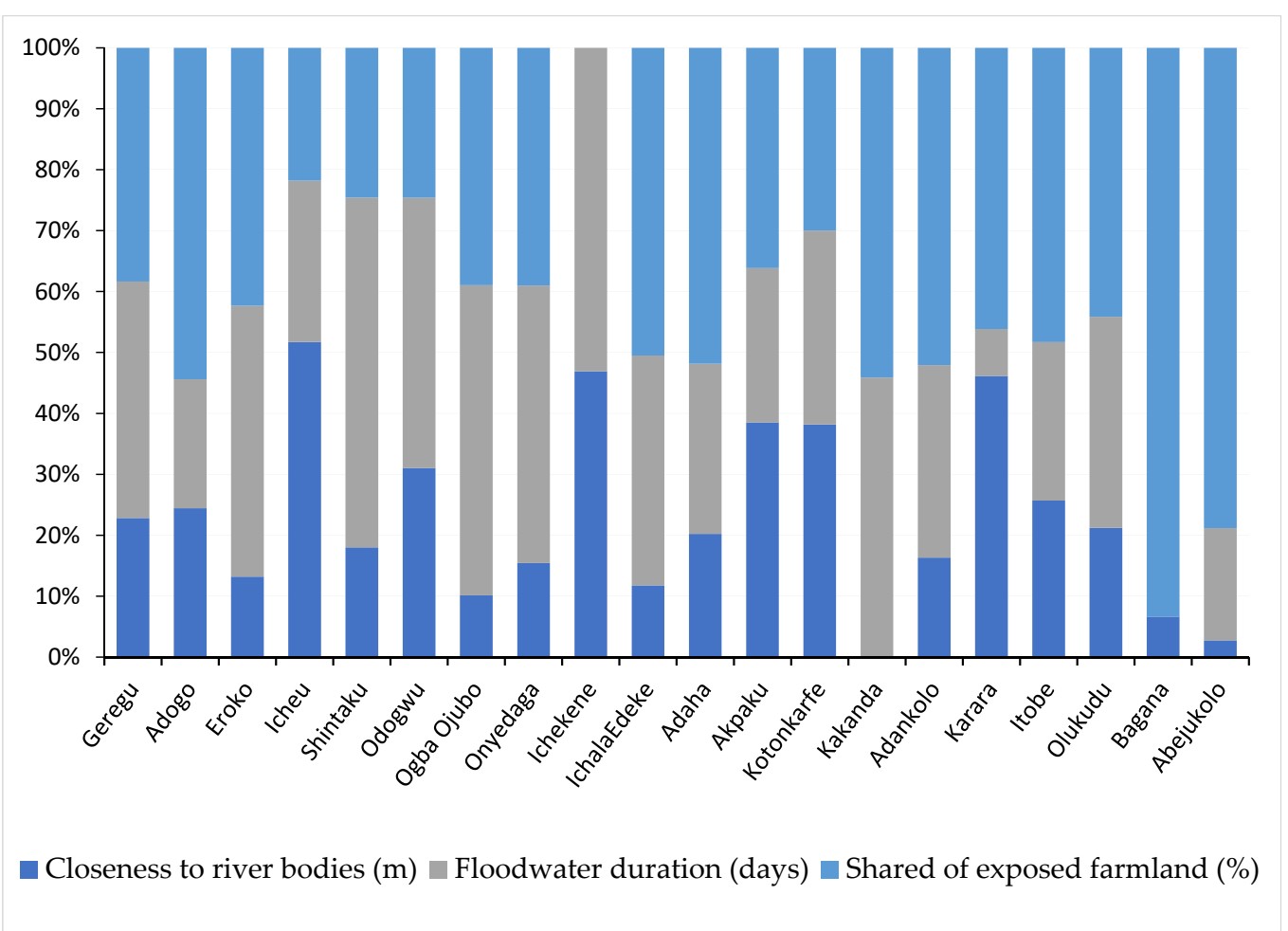

**Figure 6.** Contribution of the single indicator to the sub-index exposure (*SIE*) for the different selected flood-prone communities in Kogi State.

(b)     Contribution of the single indicator to the sub-index susceptibility (SIS) across the community.

The sub-index susceptibility is the aggregation of five indicators: household size (HS); household conditions (HCs); household past flood experience (PFE); household dependency on agriculture (HDAP); and households' lack of access to improved drinking water (LAIW). Each indicator was assessed in order to determine its contribution to susceptibility as it contributes to households' flood vulnerability. The results showed that all the indicators have significant contributions to flood susceptibility (Figure 7). In particular, four of the indicators contribute most to the current levels of flood vulnerability in all the

communities: (1) household dependency on agriculture, (2) household lack of access to improved and quality drinkable water, (3) household past flood experience, and (4) house condition. Furthermore, more than 95% of the respondents indicated high dependency on agricultural activities as their major source of income, as revealed by the survey result. The Ichala Edeke community has a sub-index value of 1.00 and was ranked the most susceptible community as a result of the contribution of all the indicators.

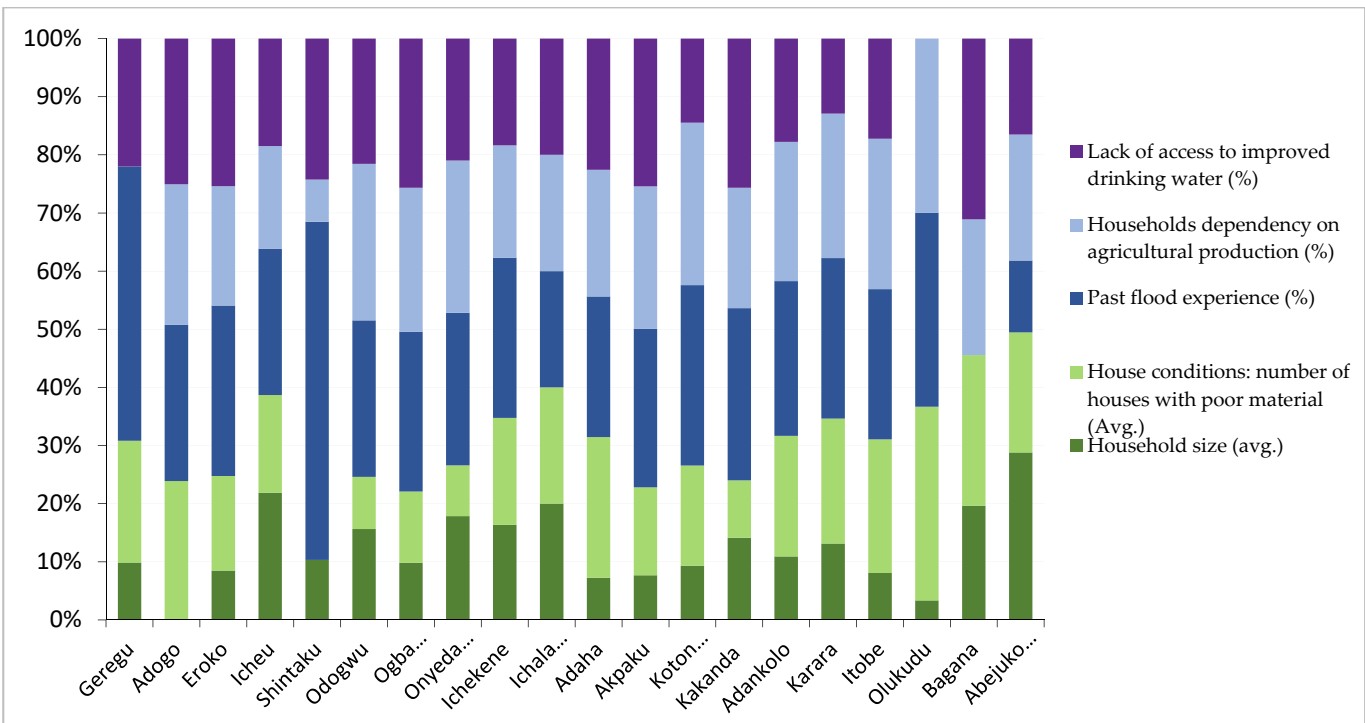

**Figure 7.** Contribution of the single indicator to the sub-index susceptibility (*SIS*) for the different selected flood-prone communities in Kogi State.

(c)     Contribution of the single indicator to sub-index lack of resilience (SILoR).

The results of the study indicated that all the surveyed communities were characterised by a lack of resilience to flooding, thereby making them more vulnerable. Figure 8 illustrates how each indicator contributes significantly to the sub-index lack of resilience across the communities, however, in a relative proportion. Specifically, five indicators contribute most to prevailing levels of lack of resilience as observed across the communities (Figure 8). They are low literacy rate, lack of access to flood management measures, inadequate financial support to recover after floods, lack of access to healthcare facilities, lack of evacuation facilities, and low flood education.

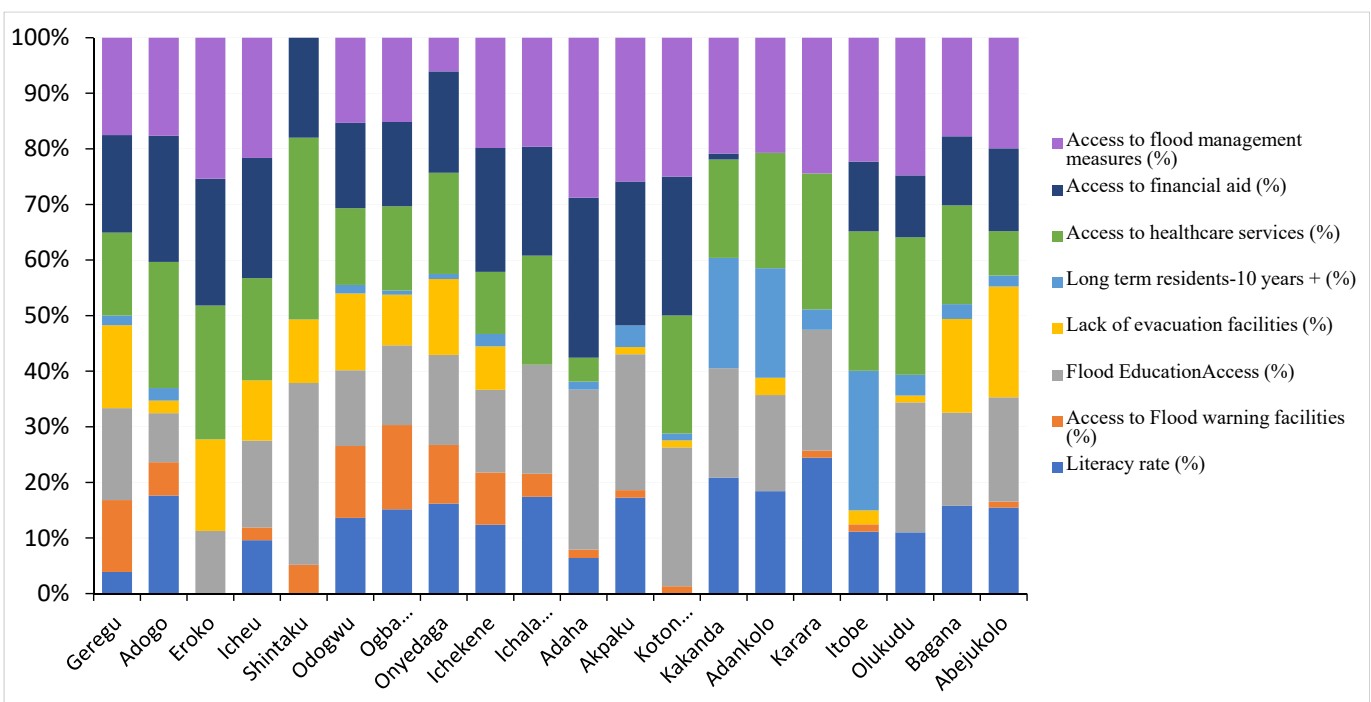

**Figure 8.** Contribution of the single indicator to the sub-index lack of resilience (*SILoR*) for the different selected flood-prone communities in Kogi State.

## 4. Discussion

A detailed understanding of the most vulnerable communities and populations, as well as the identification of the key indicators that contribute to vulnerability are essential for developing encompassing disaster risk programs, recovery plans, and policies [21,22,26,31]. Nazeer and Bork [26] noted that one of the often-employed techniques for assessing flood vulnerability is an empirical investigation using flood vulnerability composite indicators. This method is widely used in the research community to generate vulnerability indices for developing an efficient risk and flood vulnerability assessment. However, it is a relatively new method used in this study area in assessing flood vulnerability. Therefore, adopting an index-based approach, this study offers a thorough, step-by-step understanding of how people are vulnerable to flooding across the selected communities.

The MOVE and UNESCO-IHE frameworks were adopted for the description of flood vulnerability components (function of exposure, susceptibility, and resilience) and understanding indicators that best describe each component. Initially, 18 sets of indicators were initially selected through an extensive literature review, expert opinion, and field observation, with an understanding that high degree of relationship between indicators may distort the vulnerability index and mislead the end users, hence the need to discard certain highly correlated indicators [31,52]. Access to healthcare system (AHS) was found to be highly correlated with diversification of economic activity (DEA) at a correlation coefficient "r = 0.65", which is logically sound, as both of these indicators belong to building the resilience of the people in terms of recovery and coping capacity. It can be explained as the healthier the people are after flood events, the easier it will be for them to diversify their economic activities. In this case, only AHS was retained, while the DEA was discarded. Average elevation (AE) showed a strong correlation with household lack of access to improved drinking water (LAIW), suggesting an easy contamination of the waterbodies by floodwater, ao LAIW was listed among the final indicators. In the end, sixteen indicators in total were retained to construct the flood vulnerability index.

The results obtained from the analyses revealed considerable spatial variations in tract-level flood vulnerability, exposure, susceptibility, and lack of resilience across the selected

communities. In this regard, the computed FVIs lied between 0.32 and 0.74 and were used to identify the hotspots of flood vulnerability across the communities. Spatially, it was observed that Ogba Ojubo, Onyedega, Odogwu, and Ichala Edeke were the very highly vulnerable communities. A total of eleven others (Adogo, Itobe, Bagana, Akpaku, Koton-Karfe, Kankanda, Karara, Icheu, and Adankolo) had relatively high flood vulnerability, while the others fell between moderate and low flood vulnerability. Interestingly, the findings showed that all the sampled communities in Ibaji LGA (Ogba Ojubo, Onyedega, and Odogwu) had comparatively very high flood vulnerability. This result also corresponds to previous studies in Kogi State [11,30]. Similarly, a 52-year-old male member of the FGD group session at Onyedega community in Ibaji LGA said:

> *"Flooding in Ibaji LGA is always disastrous, the destruction is not limited to us of farmlands, houses but also causing serious damages and injuries to several people in this area. In fact, sometimes during flooding, people use to stay on top of trees in order to protect the life and later come down after the floodwater might have subsided".* (FGD group session, 18 June 2021)

The contribution of individual indicators to the FVI and other sub-indices so as to better understand the underlying factors contributing to flood vulnerability in the communities holistically was studied. The analyses revealed that some indicators contributed to the prevailing levels of exposure, susceptibility, and lack of resilience at varying degrees, which in turn resulted in the observed higher household vulnerability to flooding across the study area. In particular, the indicators percentage of shared flooded farmland, closeness of houses, and the longer period of days the floodwater remained in the community all contributed to the household exposure level. This conforms with the findings of Ntajal et al. [42], who found that factors such as proximity to water bodies, longer flood duration, and the location of field crops in flood zones tend to increase the exposure of communities, thus likely leading to negative impacts on humans and ecological systems. Likewise, household past flood experience, over-dependence of households on agriculture, lack of access to improved drinking water, and households' poor housing/building conditions were all identified as the main drivers of households' flood susceptibility. Here, the indicator household condition implies the number of houses with poor building materials (as noticed during the field survey) such as walls of houses made with either corrugated sheets or wooden planks, the floors of houses being bear soil and not cemented, the tops of rooves of houses being made with thatch or leaves, etc., were all found to make such households more susceptible to the impact of foods. With respect to households' over-dependence on agriculture, more than 95% of the respondents indicated high dependency on agricultural activities as their major source of income. Being largely dependent on agriculture for income may make people more vulnerable to the effects of flooding. This study result corroborated the findings of an earlier study that flooding usually has negative consequences on individuals engaging in agriculture-related activities who use agricultural lands as a source of their livelihoods [57].

Pertaining to lack of resilience, several indicators, such as households' lack of evacuation and flood management measures, low levels of flood education, high percentage of flood experience, low literacy rate, lack of access to flood warning facilities, and weak household economic capacity, were identified as the major drivers of vulnerability and lack of resilience. There is evidence in the literature that education can help increase people's resilience to flood disasters [31,47]. The results of the survey analysis showed that 85.6 % earn NGN 50,000 (equivalent of USD 120) or less per month. Of this proportion, 62.8% live below the national minimum wage of NGN 30,000. This supports the claim of high inequality in the region as indicated by a Gini coefficient of 0.64 [58,59]; with this low monthly income, the people may not be able to gather resources to prepare, anticipate, and recover from flood disasters. It is generally assumed that households with a high income or wealth are less vulnerable than those with a low income or wealth [20]. In general, these factors inhibit the household's capacity to anticipate, cope with, and recover from flooding, which supports the premise that vulnerability to flooding occurs due to households' lack of

preparedness, as shown by Ismail and Saanyol [60]. Many households depend mainly on agriculture as their major source of economic survival, causing the inhabitants to have a strong affinity for these flood-prone areas [15].

Undoubtedly, there are significant limitations to this study. The indicators' selection was largely based on the opinions of stakeholders, experts in the ministry and disaster risk management organizations, among others. Despite the fact that the datasets and indicators used in this study were considered to be the most relevant ones, methodologically, no weight was assigned to the indicators for a variety of reasons. First, we do not have the same number of indicators per sub-index. For instance, lack of resilience has more indicators than exposure and susceptibility. Secondly, no weight was assigned to the indicators in order to avoid the bias and subjectivity of the stakeholders during the fieldwork. Furthermore, as the FVI is a new approach for assessing flood vulnerability, it was a bit challenging to compare the results from the indices' components to previous studies in the area. In addition, a limited number of communities were considered for this assessment. Accordingly, a community with a high vulnerability index does not necessarily indicate that it is extremely vulnerable to floods in general, but rather, that it is much more vulnerable than the other selected communities in the study areas based on the indicators that were considered.

However, this study's logical, data-driven, and methodological soundness sets it apart from previous studies. The novel idea is not limited to identifying the comparative levels of hotspots of vulnerable communities, but also to documenting the main drivers of flood vulnerability for further analysis and actions. In essence, the results of this study can be easily applied to the local decision-making framework for flood adaptation and building resilience. In addition to the FVI approach used in this study, future research addressing vulnerability assessment at the household level should consider incorporating several indicators in disaster management plans, such as temporary relocation, insurance, communication networks, proximity to hospitals and medical care, and a flood early warning system. Due consideration should also be given to the use of GIS and remote sensing (RS) to examine the physical and anthropogenic factors contributing to flood disasters and the vulnerability of households to flooding in the region.

## 5. Conclusions

The FVI is an effective tool for flood risk reduction, recovery strategies, and policy development. It helps to gain a deep understanding of the most vulnerable communities, populations, and key indicators that truly determine the level of vulnerability of people in flood-prone areas. This paper presents the flood vulnerability index as a holistic and spatially explicit approach to assessing flood vulnerability in the selected flood-prone areas of Kogi State, Nigeria. The authors constructed the flood vulnerability index as a function of exposure to flood disasters, susceptibility to its impacts, and households' lack of resilience to anticipate, recover from, and adapt to current and future floods.

The analysis showed that households' vulnerability to flooding, exposure level, their susceptibility, and lack of resilience to the impacts of floods vary considerably across the area. The research makes three major contributions. First, it explains in detail a systematic, logical, data-driven, and methodological way of assessing flood vulnerability—the use of composite indicators to generate flood vulnerability index values for different areas, which is a new approach of assessing flood vulnerability in the region. The presented methodology can be used as a guidance tool for future flood risk and vulnerability assessments and for monitoring changes over time in the selected area and, by extension, the entire Kogi State. Secondly, the computed flood vulnerability indices' values and overall flood vulnerability maps serve as tools for identifying households in communities that are vulnerable to flooding, based on the level of exposure, susceptibility, and lack of resilience, thus facilitating the planning and prioritization of location-specific interventions for flood control. Lastly, the highlighted contributions of each indicator to the computed FVI and other sub-indices present local evidence of the issues that need to be addressed in order

to design spatial contingency plans and enable swift community/policy engagement and actions to effectively reduce households' vulnerability to flooding in the area.

**Author Contributions:** All authors were involved in the production and writing of the manuscript. Conceptualization, P.O.; data curation, P.O.; formal analysis, P.O. and Y.W.; investigation, P.O.; methodology, P.O. and Y.W.; project administration, P.O.; software, P.O.; supervision, E.K., F.O. and Y.W.; validation, P.O., E.K., F.O. and Y.W.; visualization, F.O. and Y.W.; writing—original draft, P.O.; writing—review and editing, P.O., E.K., F.O. and Y.W. All authors have read and agreed to the published version of the manuscript.

**Funding:** This research received no external funding.

**Institutional Review Board Statement:** Not applicable.

**Informed Consent Statement:** Informed consent was obtained from all subjects involved in the study.

**Data Availability Statement:** Some or all data, models, or code that support the findings of this study are available from the corresponding author upon reasonable request.

**Acknowledgments:** The authors wish to express their sincere gratitude to the West African Climate Change and Adapted Land Use (WASCAL) funded by the German Federal Ministry for Education and Research for providing financial support to the corresponding author to carry out this research as part of his postgraduate studies. Special thanks to the staff of the Environmental Vulnerability & Ecosystem Services (EVES) section, United Nations University, Institute for Environment and Human Security (UNU-EHS), Bonn, Germany, for their support and contributions to this research during the corresponding author's short scientific visit to the institution. Likewise, thanks to the technical staff of the Environment and Physical Infrastructure Policy Department of the Nigerian Institute of Social and Economic Research (NISER), Ibadan, Nigeria, for providing and enabling the working environment for the corresponding author. Finally, the authors would like to express their gratitude to the Climate Change Unit of the Kogi State Ministry of Environment and Natural Resources, as well as the Kogi State Emergency Management Agency (SEMA), Nigeria, for their collaboration in the data collection.

**Conflicts of Interest:** The authors declare no conflict of interest.

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
