# Peer review of "Understanding Flood Vulnerability in Local Communities of Kogi State, Nigeria, Using an Index-Based Approach"

_water, doi:10.3390/w14172746_

Round 1
Reviewer 1 Report
Dear authors, I give you my corrections in the attached PDF.
All the best.

Author Response
Dear Reviewer 1,
We thank you for your time and comments.
Please see the attachment.
Best regards,
Author

Reviewer 2 Report
The paper is interesting and addresses an important problem. However, there are some issues that should be addressed.
1. The paper's title could be presented more effectively.
2. Please revise the following sentence in the abstract: “Flood impacts under climate change conditions in West Africa is on the increase.”
3. In the last line of the abstract, a better contribution to practice can be stated.
4. The exposure keyword is extremely generic. Please edit it in the keywords.
5. The logic of the introduction is not acceptable. The authors can link this study to flood risk and response management strategies; please use the following papers to improve it:
· Yazdani, Maziar, et al. "An integrated decision model for managing hospital evacuation in response to an extreme flood event: A case study of the Hawkesbury‐Nepean River, NSW, Australia." Safety Science 155 (2022): 105867.
· livelihood of the local community: A case from southern Bagmati corridor of Nepal. Progress in Disaster Science, 12, 100199.
· Uddin, K., & Matin, M. A. (2021). Potential flood hazard zonation and flood shelter suitability mapping for disaster risk mitigation in Bangladesh using geospatial technology. Progress in Disaster Science, 11, 100185.
· Yildiz, A., Teeuw, R., Dickinson, J., & Roberts, J. (2021). Children's perceptions of flood risk and preparedness: A study after the May 2018 flooding in Golcuk, Turkey. Progress in Disaster Science, 9, 100143. The reference style of the paper should be revised and conform to MDPI requirements.
6. Section 2.3.2 should be formatted more effectively. The authors should thoroughly describe the Indicators and their references.
7. The discussion and the provided results should be improved.
Author Response
Dear Reviewer 2,
We thank you for your time and comments.
Please see the attachment.

Round 2
Reviewer 2 Report
Accept.